# Pathological Complete Response Following Neoadjuvant Tislelizumab Monotherapy in Treatment-Naive Locally Advanced, MMR-Deficient/MSI-High Ascending Colon Cancer: A Case Report

**DOI:** 10.3390/jcm12010240

**Published:** 2022-12-28

**Authors:** Yue Hei, Ruixia Yang, Shengnan Kong, Hongmei Zhang, Yan Chen

**Affiliations:** Department of Oncology, Xijing Hospital of Air Force Military Medical University, Xi’an 710032, China

**Keywords:** neoadjuvant immunotherapy, locally advanced colorectal cancer, tirelizumab, mismatch repair deficient, pathological complete response

## Abstract

Although recent trials started the use of neoadjuvant immunotherapy (NIT) in instability-high (MSI-H) or mismatch repair deficient (dMMR) early-stage or locally advanced colorectal cancer (LACRC), little data on the treatment strategy of NIT has been shown, and whether the tirelizumab mono-immune checkpoint inhibitor (ICI) can be used as NIT for patients with LACRC has not been reported as yet. In this study we report on a locally advanced ascending colon cancer case with a history of incomplete intestinal obstruction which achieved a pathologic complete response (pCR) after treated with Tirelizumab as NIT. A 32-year-old man was diagnosed with locally advanced ascending colon cancer with MSI-H and dMMR. An incomplete intestinal obstruction accompanied with hyperpyrexia occurred unexpectedly and was eased by symptomatic treatment. There was no peritonitis or other acute complications. NIT (three cycles of Tirelizumab) was suggested by the MDT board and partial response was achieved according to CT scanning, and pCR was further revealed by postoperative pathology. A ctDNA clearance confirmed the R0 resection and some immunotherapy related predictors were also detected using the NGS method. Our case study contributes to the evidence on the feasibility, efficacy, and safety of f Tirelizumab as a mono ICI for an optional neoadjuvant therapy in patients with MSI-H/dMMR LACRC.

## 1. Introduction

Although microsatellite instability-high (MSI-H) or mismatch repair deficient (dMMR) tumors appear to be less responsive to neoadjuvant chemotherapy, based on the successful outcomes of KEYNOTE-016 and CheckMate-142 studies in metastatic colorectal cancers (mCRC), immunotherapy was approved for standard first-line treatment for MSI-H/dMMR mCRC [1]. Recently, with flourishing trials such as NICHE starting the use of neoadjuvant immunotherapy (NIT), it is expected to be more widely used in early-stage or locally advanced colorectal cancer (LACRC, defined as CRC stage II (cT3–4, N0)/stage III (any cT, N+)) in the future [2,3]. As is known, the tumor microenvironment undergoes immunosuppressive changes with disease progression, and early intervention of NIT could increase the activity of anti-tumor immune T cells and reverse this situation, which is helpful in eliminating small metastatic lesions and achieving a clinical complete response (cCR) or even a pathological complete response (pCR) [4]. However, little data on the treatment strategy and evaluation of NIT has been shown for LACRC, and many disputes exist.

Compared with the difficulties faced, the opportunities far outweigh the challenges. A prospective phase 2 study demonstrated that dostarlimab monotherapy for 6 months could result in a cCR rate of 100% in patients with LACRC [5], indicating its high response to the single-agent immune checkpoint inhibitor (ICI). It is likely that tirelizumab (BGB-A317) is a humanized IgG4 anti-programmed death 1 (anti-PD-1) monoclonal antibody, and its current National Medical Products Administration (NMPA) indication for MSI-H/dMMR solid tumors was approved based on a clinical trial of a phase 2 study in patients with previously-treated locally advanced unresectable or metastatic MSI-H/dMMR solid tumors, which included 46 patients (62.2%) with advanced CRC. The objective response rate (ORR) was 45.9% in total, and the ORR was 39.1% in CRC patients [6]. A recent publication has shown that Tirelizumab plus chemoradiation (CCRT) should be one option of neoadjuvant therapies for locally advanced rectal cancers with dMMR/MSI-H status [7]. However, whether Tirelizumab mono-ICI can be used as NIT for patients with LACRC has yet to be reported. Here we report on a 32-year-old man with locally advanced ascending colon cancer who underwent first-line Tirelizumab monotherapy and achieved pCR according to the pathological assessment after surgery.

## 2. Case Presentation

A 32-year-old man underwent laparoscopic appendectomy due to appendicitis in the local hospital in June 2022 and a neoplasm was found to be occupying the ascending colon. A colonoscopy indicated a huge annular mucosal ulcer in the ileocecal region covered with dirty moss. Computed tomography (CT) scanning of the chest and abdomen showed wall thickening of the ascending colon with multiple peripheral enlarged lymph nodes that were considered to be malignant (Figure 1B). A biopsy suggested a highly differentiated adenocarcinoma (Figure 2A,B), and immunohistochemistry further showed that *MLH1* (−), *MSH2* (+), *MSH6* (+), *PMS2* (−), *Ki-67* (+60%), *PD-L1* (22C3, CPS) (+60%), *EGFR* (+), *KRAS*, *NRAS*, and *BRAF* gene mutations were not detected. Microsatellite instability (MSI)-H was detected by NGS. Relevant past history included smoking an average of 20 cigarettes per day for more than 10 years. Thus, the patient was diagnosed with adenocarcinoma of the ascending colon (cT3N2, stage III).

Unexpectedly, abdominal pain accompanied with hyperpyrexia occurred on 22 June 2022. A dual-source abdominal CT showed multiple gas accumulation in the intestinal cavity, with a diagnosis of incomplete intestinal obstruction (Figure 1A). After gastrointestinal decompression and symptomatic treatment, the abdominal pain was eased, and laboratory markings of inflammatory response such as C-reactive protein and procalcitonin decreased significantly. In consideration of no indication of peritonitis or other acute complications, neoadjuvant treatment was recommended by MDT discussion. In this particular case, the ECOG PS score of the patient was relatively poor and was not suitable for intensive treatment, and ICI plus chemotherapy or double-ICIs (anti-CTLA4 plus anti-PD-1) retrieved no data. As tirelizumab’s current National Medical Products Administration (NMPA) indication for MSI-H/dMMR solid tumors was approved on 11 March 2022 and there are some studies referring to anti-PD-1 monotherapy (including pembrolizumab, sintilimab and nivolumab), tirelizumab was used for NIT treatment in this case.

The patient underwent three cycles of NIT with tirelizumab 200 mg ivgtt. Q3W from 2022-06, partial response was achieved after two cycles of treatment according to the criteria of RECISTv1.1 (significant regression for ileocecal mass and peripheral lymph nodes). The laparoscopic radical resection of the colon cancer was conducted in September 2022. Intraoperative exploration showed that the ileocecal mass was non-invasive to the peritoneum. Postoperative pathology further revealed pCR with severe inflammation of the colon mucosa with erosion, bleeding and ulcer formation (Figure 2C,D). No cancer cells were found at the bilateral surgical margins. Neither mesenteric lymph nodes (0/1) or lymph nodes around the right colonic artery (0/23) were found with metastatic cancer. TRG is classified as Grade 0. As suggested by the Department of Gastrointestinal Surgery, a colonoscopy was performed one year post-surgery to check the local conditions of the patient. A further liquid biopsy, using circulating tumor DNA (ctDNA) and minimal residual disease (MRD) detection, turned out to be negative (ctDNA clearance), which confirmed the R0 resection, and some immunotherapy-related predictors were also detected using the next-generation sequencing (NGS) method (Table 1). Other gene mutations of potential clinical significance are shown in Appendix A. We used the CARE checklist when writing our report (See detailed information in Appendix A).

## 3. Discussion

It is generally believed that patients with MSI-H/dMMR LACRC have poor responses to 5-fluorouracil (5-FU)-based neoadjuvant chemotherapy [8]. However, studies such as that of Cercek et al. [9] demonstrated that 93% (13/16) dMMR patients with rectal cancer experienced tumor downstaging after neoadjuvant CCRT. Nevertheless, NIT for patients with MSI-H/dMMR LACRC is still of huge significance and is beneficial for organ preservation [10]. We presented here a case of locally advanced ascending colon cancer with incomplete intestinal obstruction that was treated with Tirelizumab alone as the NIT regime, and we achieved a pCR after surgery, which further indicated the feasibility, efficacy, and safety of Tirelizumab as a mono-ICI for an optional neoadjuvant therapy in patients with MSI-H/dMMR LACRC.

Given the patient’s individual differences and the variety of NIT treatment strategies, the appropriate treatment settings, including the treatment duration and best consolidation therapy after pCR, has not been determined. Previous case studies suggested that pCR could be achieved using both short-course NIT (ipilimumab once plus nivolumab twice) or long-term monotherapy (nivolumab for six cycles) [10,11]. Zhang et al. [12], reported 32 patients of LACRC using a median six cycles of ICI from a retrospective study, and the ORR was shown to be 100% and the rate of major pathological response (MPR) was 86.2% in terms of TRG. Prolonged waiting time spent in NIT may not change the pCR rate and is associated with a higher incidence of disease progression and more challenging surgical resection. In the present study, we used three cycles of NIT, which is a moderate time course in comparison to the previous studies. Furthermore, the detection of liquid biopsy and NGS data was done later, and covered 437 exons, fusion related introns, variable splicing regions and specific microsatellite site regions. The results of the detection include point mutation, small fragment insertion deletion mutation, gene fusion and copy number variation within the coverage, most of which has no clear guiding effect on clinical practice. Instead, ctDNA clearance has obvious clinical significance and takes an important role in indicating tumor burden and risk stratification. We detected ctDNA, but its lack of supporting information how to use it in immune-consolidation therapy. Recent studies have shown the ability of ctDNA in prognosis prediction in neoadjuvant therapy-treated LACRC patients: patients with MRD who received post-surgery ICI treatment had significantly better outcomes than the patients who did not; in contrast, the prognosis of patients without MRD was better whether they received ICI or not [13]. Although how to choose between “watch and wait” and immune-consolidation therapy requires further evidence in patients with ctDNA clearance, it seems that prolonged ctDNA detection may facilitate the individualization of consolidation therapy.

Gastrointestinal (GI) toxicities could be induced by ICI treatment alone and the most common symptoms are diarrhea and colitis, which may result in further treatment discontinuation or even intestinal obstruction [14]. In this case, the patient experienced an incomplete intestinal obstruction accompanied with hyperpyrexia just before systemic treatment, and therefore he is prone to generate GI related events and was not suitable for intensive treatment, and thus mono-ICI treatment was given and the gastrointestinal condition and therapeutic response was closely monitored. It is notable that, different from chemoradiotherapy and targeted therapy, one of the characteristics of NIT is that the evaluation results between the imaging and pathology may differ greatly [15]. Specifically, in this case, the tumor size, instead of cCR on imaging, as expected, merely became smaller and the tumor mass was still evident due to the infiltration of immune cells proven to be necrotic by postoperative pathology. Thus, this further demonstrated the limitation of traditional image assessment in terms of treatment response.

Furthermore, recent studies tried to find the safety and efficacy of neoadjuvant chemoradiotherapy plus tislelizumab followed by total mesorectal excision for LACRC [16]. If the rational neoadjuvant therapy could help overcome the immunosuppressive tumor microenvironment (IL-γ changes, DC cell maturation/activation, M2 macrophages increasement, and the recruitment and stimulation of CD8+T cells, etc), these combined strategies with immunotherapy have the possibility to start the potential anti-tumor immune response [17]. Taken together, enhancing anti-tumor efficacy with immunotherapy combinations still has many uncertainties and needs more clinical trials for verification.

MMR status was found to affect the pCR rate after neoadjuvant therapies in LACRC, though the outcomes were controversial (positive or negative or even insignificant effects) [15,18,19]. As a matter of fact, there is a discordance in the detection methods, results interpretation and technical platforms, as well as the heterogeneity of tumors, which may partly explain the unsatisfactory prediction of the therapeutic effect of immunotherapy using MMR. Previous studies also identified that cN0 and the clinical T stage were related to pCR [18]. Taking into consideration that MMR status may not precisely predict the immunotherapeutic value alone, other therapeutic predictive markers such as *TMB* and *POLE* and *POLD1* gene mutations are required and should be interpreted comprehensively [20]. There are still many disputes about the threshold of TMB and the utility of programmed death ligand-1 (PD-L1) has yet to be validated, as it could be affected during treatment. *POLE* and *POLD1* gene mutations can be used as independent predictors both for the efficacy of pan-cancer immunotherapy and poor prognosis. Recent studies further presented stromal PD-L1^+^ immune cells and nuclear β-catenin^+^ tumor budding as a combinative biomarker for tumor progression and treatment resistance in LACRC [21]. Finally, even though we used individualized methods (as above) for prognosis prediction, it should be pointed out that it remains disputable whether pCR after NIT contributes to longer overall survival, other than disease-free survival in LACRC patients. All in all, it is necessary to dig deeper into the molecular structure of tumors and search for a comprehensive prognosis predictor and individualized treatment.

## 4. Conclusions

In the present case, we show the feasibility, efficacy, and safety of tirelizumab as a mono-ICI for an optional neoadjuvant therapy in treatment-naive MSI-H/dMMR LACRC patients, even those with a history of incomplete intestinal obstruction. The novelty of the study lies in: (1) the use of tirelizumab as a mono-ICI for NIT therapy in dMMR/MSI-H locally advanced ascending colon cancer; (2) its use even in those with a history of incomplete intestinal obstruction; and (3) few reports of intensive study through various methods (liquid biopsy, NGS data et al.) were shown in this area. Our study provides evidence for the adjunctive treatment of dMMR locally advanced CRC with tiralizumab. As a matter of fact, although NIT holds curative promise, further clinical data were required for the use of ctDNA MRD detection, and it is recommended that researchers dig deep into the molecular structure of tumors and search for a more comprehensive prognosis predictor.

## Figures and Tables

**Figure 1 jcm-12-00240-f001:**
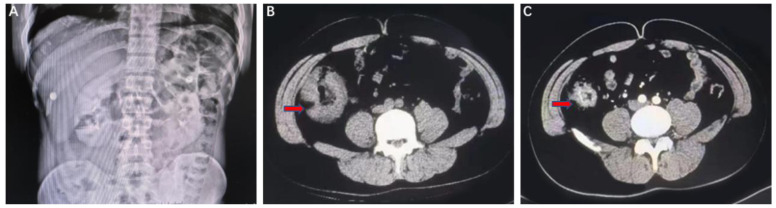
Imaging evaluation before and after treatment. Multiple gas accumulation in the intestinal cavity before treatment shown by CT scanning (**A**); Tumor mass in the ascending colon as indicated by red arrows (**B**), and significantly reduced after three cycles of immunotherapy (**C**).

**Figure 2 jcm-12-00240-f002:**
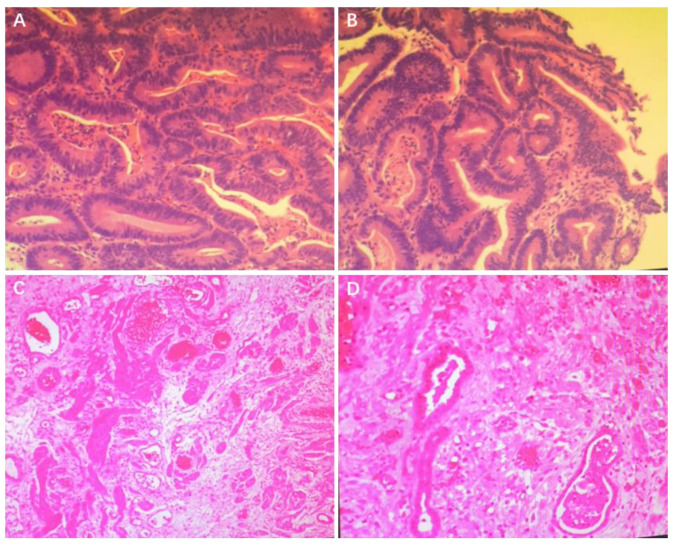
Histopathological findings before and after treatment. The biopsy before treatment turned out to be a highly differentiated adenocarcinoma (**A**,**B**). A large inflammatory cell infiltration, interstitial congestion, hemorrhage, and local fibroplasia could be seen without any malignant evidence after surgery (×40 magnification) (**C**,**D**).

**Table 1 jcm-12-00240-t001:** Detection of immunotherapy related predictors.

Immuno-positive	*PMS2* mutation	*POLE* not detected	*POLD1* not detected
Immuno-negative	*B2M* mutation	*JAK1* mutation	*STK11* not detected
Hyperprogression	None		
TMB	65.9 Muts/Mb		
MSI-H	*PMS2* mutation		

## Data Availability

The original contributions presented in the study are included in the article. Further inquiries can be directed to the corresponding authors.

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
