# Peer review of "Pathological Complete Response Following Neoadjuvant Tislelizumab Monotherapy in Treatment-Naive Locally Advanced, MMR-Deficient/MSI-High Ascending Colon Cancer: A Case Report"

_jcm, 2022, doi:10.3390/jcm12010240_

Round 1

Reviewer 1 Report

The article presents a new method of neoadjuvant treatment in non-obstructive colon cancer. The result is favorable and demonstrates the effectiveness of this treatment. Being only a case presentation, no general conclusions can be drawn, but a study on several cases can be edifying. I suggest the authors to continue the research. The language is appropriate and the bibliography is edifying. I consider that the case presentation meets the conditions for publication

Reviewer 2 Report

This is a very interesting case report with sufficient methodology and results.The authors have successfully described the case of the locally advanced ascending colon cancer, and the language is very concise. I have 4 minor suggestions for this manuscript to be improved and be able to be published.

1. Please fill in and include CARE reporting checklist.

2. The use of liquid biopsy and NGS data for the detection of MRD is very interesting and crucial for the overall study. The authors should discuss more their findings in the results and discussion section as well as find a better way to present better their findings (e.g. a figure of NGS data) rather than only a table.

3. As this is not the first case report describing Tislelizumab monotherapy in locally advanced  ascending colon cancer, the authors should better clarify the novelty of their study in the discussion or conclusion section.

4. I suggest that you remove study's limitations  from the conclusions and further support your findings in the conclusion. The authors should further discuss the limitations of their case report in the discussion section.

Reviewer 3 Report

Authors presented a colorectal cancer (CRC) case treated with neoadjuvant tiselizumab. The following issues should be resolved.

1. The presentation of the case is poor. Further details are required, e.g., provide further details why only diagnostic laparoscopy was performed. Furthermore, more emphasis is needed why tiselizumab was choosen as the most fitting neoadjuvant option.

Although some additional information could be found in discussion, but the presentation of the case should be as complete as possible in the "Results" section first. Readers should also full understand "the story" behind each step.

2. Both the introduction and discussion is rough. Thoughts do not always follow each other logically. For example, one moment authors write about ICIs in general, and then in the next sentence a never-before-mentioned zumab is there, which is not even what the patient received. Why? PD-1, PD-L1, everything appears randomly.

3. Figure 1: Please, put (red) arrows to highlight the differences for non-radiologists.

4. English should be chekced during revisions.

Round 2

Reviewer 3 Report

The revised manuscript improved significantly.